# EXPLAINING PATTERNS IN DATA WITH LANGUAGE MODELS VIA INTERPRETABLE AUTOPROMPTING

## ABSTRACT

Large language models (LLMs) have displayed an impressive ability to harness natural language to perform complex tasks. In this work, we explore whether we can leverage this learned ability to find and explain patterns in data. Specifically, given a pre-trained LLM and data examples, we introduce interpretable autoprompting (iPrompt), an algorithm that generates a natural-language string explaining the data. iPrompt iteratively alternates between generating explanations with an LLM and reranking them based on their performance when used as a prompt. Experiments on a wide range of datasets, from synthetic mathematics to natural-language understanding, show that iPrompt can yield meaningful insights by accurately finding groundtruth dataset descriptions. Moreover, the prompts produced by iPrompt are simultaneously human-interpretable and highly effective for generalization: on real-world sentiment classification datasets, iPrompt produces prompts that match or even improve upon human-written prompts for GPT-3. Finally, experiments with an fMRI dataset show the potential for iPrompt to aid in scientific discovery.

## 1 INTRODUCTION

Large language models (LLMs) have attained an extraordinary ability to harness natural language for solving diverse natural-language problems (Devlin et al., 2018), often without the need for fine-tuning (Brown et al., 2020; Sanh et al., 2021). Moreover, LLMs have demonstrated the capacity to excel at real-world problems, such as mathematics (Lewkowycz et al., 2022) and scientific question answering (Sadat & Caragea, 2022).

In this work, we probe whether we can leverage the learned skills of an LLM to *find and explain patterns* in a dataset. To do so, we invert the typical problem of fitting an LLM to data and instead ask whether we can use a fixed LLM to produce a natural-language string explaining dataset patterns. Our approach to this problem centers around prompting. Prompting has emerged as an effective method for adapting LLMs to perform new tasks (Liu et al., 2021a). A prompt string is combined with each example in a dataset before querying an LLM for an answer.

While prompts were initially constructed manually, recent work has shown success in *autoprompting*, i.e. automatically finding a prompt via optimization (Shin et al., 2020; Li & Liang, 2021). However, previous work on learning natural language prompts Shin et al. (2020) does not produce prompts that are meaningful to humans.

Our approach, interpretable autoprompting (*iPrompt*), extends autoprompting to generate a semantically meaningful natural-language prompt that explains a key characteristic of the data (see Fig. 3). For example, given a dataset of examples of addition, e.g. $2\ \ 5 \Rightarrow 7 \ ... \ 3\ \ 1 \Rightarrow 4$, we use an LLM to yield the natural-language description *Add the inputs*. iPrompt is an iterative algorithm that alternates between (i) proposing candidate explanations with an LLM, (ii) reranking the candidates based on their performance when used as a prompt, and (iii) exploring new candidates.

To evaluate iPrompt, we curate a diverse collection of datasets written in natural language (Table 1), where our goal is to accurately infer a ground-truth pattern. The dataset includes a number of synthetic math datasets, as well as language tasks from the Natural Instructions V2 dataset (Wang et al., 2022). We find that iPrompt outperforms baseline autoprompting methods in successfully finding a correct description across these datasets. Moreover, the generated descriptions are interpretable,

Figure 1: Interpretable autoprompting (*iPrompt*) inverts the standard prediction problem to instead find a natural-language explanation of the data using a fixed, pre-trained large language model (LLM).

Table 1: Dataset Explanation Task. For full details on each dataset, see Appendix A.1.

| Collection | # | Description | Dataset names |
|---|---|---|---|
| Inverse synthetic math | 10 | Simple mathematical functions | Add two, Subtract two, Multiply two, Divide two, Max two, First number, Square, Exponentiate, Double, Fibonacci |
| Inverse Allen NLI (Wang et al., 2022) | 10 | Diverse language tasks | Country capital, Antonyms, Check edibility, Rhyme generation, Country currency, Check prime, Check vegetarian, Find typo, Gender classification, SQL query generation |
| Sentiment | 4 | Sentiment classification | SST-2, RottenTomatoes, IMDB, Financial Phrasebank |
| Natural-language fMRI (Huth et al., 2016) | 20 | Find an underlying category from a list of words that excite an fMRI voxel | Extracting a pattern from a set of words, each corresponding to a different voxel |

allowing human auditing and enabling strong generalization performance when used as a prompt in a new setting (i.e. when used for a different LLM). On real-world sentiment classification datasets, iPrompt even produces prompts that match or improve upon human-written prompts for GPT-3. Finally, we qualitatively explore iPrompt in a neuroscience task, in which we seek to understand the mapping of semantic concepts in the brain from fMRI imaging (data from Huth et al. (2016)).

## 2 DATASET EXPLANATION TASK

**Task definition** Given a dataset comprised of input-output string pairs $\{(x^1, y^1), \ldots (x^N, y^N)\}$, the goal is to produce a "semantically meaningful" natural-language string that explains the relationship between $x$ and $y$. We require that a string consists of human-understandable text rather than a sequence of incongruous tokens. For example, in the task shown in Fig. 3, the task is to recover text synonymous to *Add the inputs* given samples of data performing addition.

**Datasets** Table 1 shows the four collections of datasets we study: (1) Inverse Synthetic Math with datasets that require inferring an underlying mathematical function of one or two numbers; (2) Inverse Allen NLI (ANLI), a selection of crowdsourced language tasks (Wang et al., 2022) with easily verifiable descriptions (e.g. *Find a country's capital*); (3) Sentiment, consisting of four real-world sentiment classification tasks and (4) fMRI, a dataset involving brain responses to natural language, motivated by the goal of recovering unknown explanations. In addition to data examples, the first two collections contain a ground-truth description and simple rules to test whether an extracted description matches the ground-truth one. For example, when adding two numbers (Fig. 3), the rule checks whether a description contains any of the keywords *add*, *sum*, or *+*.

The examples in each task do not directly contain the task description. For example, when inferring the *Add two numbers* task, the examples do not contain a plus sign or any synonyms of the word *add* such as *combine*. For classification tasks such as *Check edibility* or *Check prime*, the label provided in the example text is simply *yes/no* rather than the given labels, e.g. *edible/non-edible*.

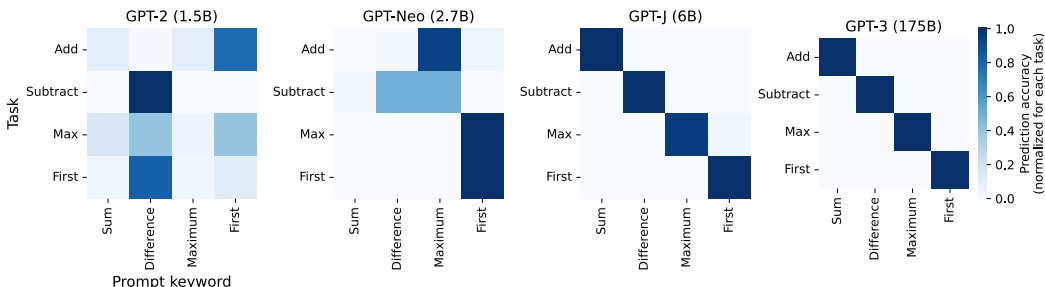

Figure 2: Model accuracy depends on having an accurate prompt for large models (GPT-J 6B and GPT-3). The model is given the prompt *Return the ___of the inputs.*, where ___ is filled in with the shown prompt keyword before querying the output given two inputs numbers in a string. Darker indicates a higher accuracy, and high accuracy along the diagonal indicates that the correct prompt induces the highest accuracy.

**Evaluation**   We evaluate dataset explanation on two criteria: closeness to the ground-truth prompt and ability to generalize as a prompt for other models. To evaluate similarity to the ground truth, we score a ranked list of prompts based on mean reciprocal rank (MRR). Given a set of datasets $\mathcal{D} = \{\mathcal{D}_1, ..., \mathcal{D}_N\}$, we compute: MRR $= \frac{1}{|\mathcal{D}|} \sum_{i=1}^{|\mathcal{D}|} \frac{1}{\text{rank}_i}$, where $\text{rank}_i$ is the one-indexed rank of the first correct explanation. We evaluate correctness based on whether the generated explanation contains one of a set of problem-specific keywords. To measure generalization, we use the top-ranked string as a zero-shot prompt for a different language model, and evaluate whether that model is able to solve the task.

## 3   AUTOPROMPTING METHODS

In this section, we detail approaches for tackling the general problem of autoprompting before introducing our method for interpretable autoprompting (iPrompt) in Sec. 3.2.

We specify autoprompting as a discrete search problem. Given a dataset of $n$ input-output pairs $\{(x^1, y^1), ..., (x^n, y^n)\}$ and a pre-trained LLM $f$ that returns the log-probability of a given string, the goal of autoprompting is to find a natural-language explanation $\hat{s}$ maximizing:

$$\hat{s} = \operatorname*{argmax}_{s \in \mathcal{S}} \sum_{i=1}^{n} f\left(\text{render}(s, x^i, y^i)\right) \tag{1}$$

The render function is a problem-specific function that renders a natural language string from the prompt $s$ and each example in the dataset $(x^i, y^i)$. We use $\mathcal{S}$ to indicate the set of fluent strings, under some notion of syntactic fluency. Solving this search problem exactly is intractable.

A core assumption of this objective is that semantically accurate prompts lead a model to assign higher probability to the data. To check this assumption, we analyze four datasets from the inverse synthetic math collection that share common structure for the inputs and prompts: each dataset admits a prompt of the form *Return the ___ of the inputs.*, then is given two input numbers and queried for the output.

Fig. 2 shows the accuracy of different models at performing these tasks when given different input prompts.[1] For small models, the prompts are unsuccessful, but for large models (e.g. GPT-J 6B and GPT-3), the model is accurate if and only if given the correct prompt.[2] This result provides evidence that, at least for large models, the search for a prompt that maximizes performance correlates well with the underlying task.

---

[1]The accuracy is normalized for each task using softmax in order to visualize the effect of differing prompts.
[2]For details on each model, see Table A3.

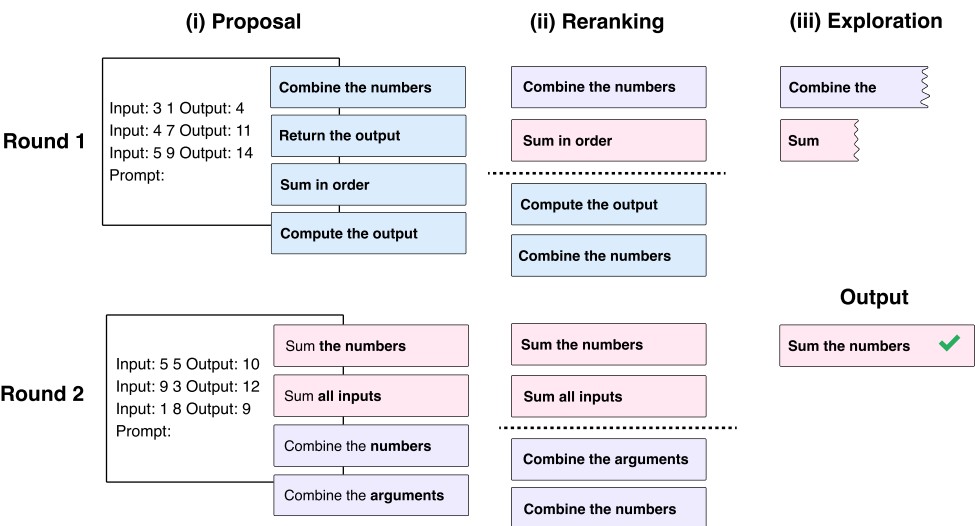

Figure 3: Overview of iPrompt.

## 3.1 BASELINE METHODS

**AutoPrompt** AutoPrompt (Shin et al., 2020) targets the objective posed in Eq. (1) using a gradient-based local search. AutoPrompt searches for $\hat{s}$ following the gradients of the objective Eq. (1) with respect to individual tokens in $\hat{s}$. By iteratively computing these gradients, it can discretely change individual words in $\hat{s}$ and then check whether or not the newly updated $\hat{s}$ improves the objective score. The use of gradients allows AutoPrompt to find an effective prompt $\hat{s}$, but makes it difficult to find answers that satisfy the fluency constraint $\mathcal{S}$.

**Average-output suffix decoding** LLMs themselves can be directly used to predict prompt strings. We can give the model a prompt that includes examples such as the following context string: $\underbrace{In: 2\,5}_{x^i}\underbrace{Out:\,7.}_{y^i}\underbrace{To\ compute\ the\ output\ from\ the\ input,}_{template}$ ___, and sample the output for the blank to recover a prompt $\hat{s}$. Sampling directly from $f$ helps ensure that the generated explanation is fluent and semantically meaningful. We decode the output using beam search to find the highest-probability outputs for multi-token prompts.[3] To improve on this approach, we place several examples into the model's context, and then average the model's output logits across all the examples in the dataset before decoding the output, an approach we refer to as *average-suffix decoding*. However, this technique is insufficient to find high-scoring prompts.

## 3.2 PROPOSED METHOD: IPROMPT

Our method, iPrompt shown in Fig. 3, is a iterative local search algorithm that alternates between three steps: (i) proposing candidate prompts, (ii) reranking candidate prompts, (iii) exploration:

(i) Proposal: Candidate prompts are generated by extending the zero-shot LLM generation. Given a data instance as a prefix, we sample a number of candidate prompts.[4] The maximum length of each candidate is pre-specified and fixed. For example, in the add-two-numbers task (Fig. 3), we may generate four candidates: {*Combine the numbers, Return the output, Sum in order, Compute the output*}.

---

[3]Here we prefer beam search here over alternatives such as nucleus sampling (Holtzman et al., 2019) as we are interested in finding an accurate prompt description with as few samples as possible.

[4]One could use either average suffix decoding or suffix decoding with a single sample. For computational efficiency, we use suffix decoding with only a single sample. We also add randomly decode the output rather than using beam-search, as our iterative procedure can recover from initially finding inaccurate candidates.

(ii) Reranking: Given candidates, the objective Eq. (1) is evaluated for each candidate prompt $s$. The top few candidates which maximize the objective are kept, e.g. narrowing down the candidates to {*Combine the numbers, Sum in order*}.

(iii) Exploration: Each of the top candidates from reranking is truncated at a random position. These truncated candidates are used as a prefix string when generating new candidate prompts via suffix decoding. For example, we may randomly select the start of the previous candidates and fill in the endings: {*Combine the ___, Sum ___*} → {*Combine the numbers, Combine both arguments, Sum the numbers, Sum all inputs*}

The algorithm is repeated until identifying a suitably strong $\hat{s}$, e.g. selecting *Sum the numbers*. Step (i) and (iii) ensure that prompts remain fluent, while step (ii) improves the score of the prompts on the objective. Computationally, iPrompt only requires running inference on the pre-trained LLM, yielding a significantly lower memory requirement than methods such as AutoPrompt, which require access to the LLM's gradients.

## 4 RESULTS

**Accuracy of prompts**   Table 2 compares prompting methods based on the set of candidate descriptions they generate using GPT-J (a 6-billion parameter model) as the LLM (Wang & Komatsuzaki, 2021). The MRR rows show that iPrompt considerably increases the mean reciprocal rank (MRR) (Sec. 2) over the baselines, implying that iPrompt can more effectively generate descriptions that accurately reflect the underlying data pattern. The "top-prompt correctness" rows show the percentage of datasets for which the top-ranked candidate prompt produced by each method is labeled as accurate by manual inspection (see all prompts in Appendix A.2). On the ANLI datasets, iPrompt again outperforms the baselines, although all methods perform poor in an absolute sense ($\leq 30\%$). The zero-shot results show the accuracy of GPT-J when using the top prompt found by each model; for the math datasets the iPrompt prompt elicits an improvement over the baselines, but for the ANLI datasets all prompts induce poor performance.[5]

Table 2: Accuracy for dataset explanation measured via (i) MRR, (ii) top-prompt correctness, and (iii) zero-shot accuracy on unseen examples. All experiments are on GPT-J 6B. For all metrics, higher is better.

|      |                        | **iPrompt** | AutoPrompt | Average suffix |
|------|------------------------|-------------|------------|----------------|
| Math | MRR                    | **0.71**    | 0.30       | 0.07           |
|      | Top-prompt correctness | **80%**     | 30%        | 20%            |
|      | Zero-shot acc.         | **51.5%**   | 41.6%      | 10.0%          |
| ANLI | MRR                    | **0.30**    | 0.17       | 0.01           |
|      | Top-prompt correctness | **30%**     | 0%         | 10%            |
|      | Zero-shot acc.         | 4.7%        | 1.9%       | **5.1%**       |

**Qualitative assessment of top prompts**   Table 4 shows the top-ranked prompt generated by each method for selected datasets. iPrompt often finds a prompt that is somewhat indicative of the underlying relationship, even if the phrasing is not perfect. For example, for the *add two numbers* dataset, it finds *Write a function int add(*. For difficult datasets, the iPrompt string sometimes simply returns the classes of the output (e.g. *yes or no?*) rather than capturing the underlying relationship. The prompts found by iPrompt also read as coherent strings compared to AutoPrompt, which returns an incoherent set of tokens. See all found prompts, including for average-suffix decoding in Appendix A.2.

**Generalization of generated prompts to new models.**   Table 3 shows the generalization accuracy when using the prompts generated using GPT-J (Table 4) and testing them on different LLMs.[6] The

---

[5]Here, we restrict generated prompts to 6 tokens (see a full discussion of experimental details in Appendix A.3).

[6]Accuracy is computed following Brown et al. (2020); Raffel et al. (2020): using exact matching with beam search, a beam width of 4, and a length penalty of $\alpha = 0.6$.

Table 3: Generalization accuracy (zero-shot) when testing the prompts generated with GPT-J as the LLM across different models. iPrompt yields strong performance, usually improving over Auto-Prompt despite maintaining interpretability, and sometimes performing close to the human-written prompt. Numbers within 2% of the top accuracy (excluding human-written prompts) for each model are shown in bold.

|  |  | Human-written | **iPrompt** | AutoPrompt | Average Suffix | No prompt |
|---|---|---|---|---|---|---|
| Math | OPT 6.7B | 12.7 | **19.3** | **18.9** | 4.5 | 8.4 |
|  | GPT 20B | 76.1 | **54.4** | 23.2 | 21.3 | 8.5 |
|  | GPT-3 175B | 76.0 | **62.1** | 40.8 | 16.9 | 28.4 |
| ANLI | OPT 6.7B | 10.7 | **6.7** | 4.7 | **8.5** | **7.9** |
|  | GPT 20B | 31.0 | **14.2** | 5.6 | **13.7** | 4.0 |
|  | GPT-3 175B | 37.6 | **11.7** | 2.7 | **13.4** | 7.7 |

Table 4: Examples of generated prompts.

|  | > Human-written prompt | **iPrompt** | AutoPrompt |
|---|---|---|---|
| ANLI | > Generate an SQL statement from a question asking for certain data. | Write an SQL to produce output | ributed grandfatherExceptionappropri intent Lara |
|  | > You are given a country name and you need to return the currency of the given country. | Select currency code for a new | renciesthethe Dmitrythe mortg |
|  | > Return whether the input food dish is vegetarian (yes or no). | yes or no? This is | Novthethethethethe |
|  | > In this task, you are given an adjective, and your job is to generate its antonym. An antonym of a word is a word opposite in meaning to it. | What is the opposite of 1 | prevailingthethe weakestthe wins |
| Math | > Return the sum of the inputs. | Write a function int add( | addedthe +the use worked |
|  | > Return the product of the inputs. | When you multiply two ( | multiplythethe the Multiple |
|  | > Return the difference of the inputs. | If n > m then subtract | opposably exactly subtractFor YEAR |
|  | > Return the maximum of the inputs. | Which number has a bigger value | NumberthetheJusticeJaDefault |
|  | > Return the first of the inputs. | The first digit of both values | greater name sorting indiscrim to numbers |
|  | > Square the input to get the output. | Write a function that calculates square | multiplythe hypot Norttheirl |
|  | > Given an input x, return 2*x. | write a function called double that | ADDthe introducedpareat contraceptives |
| Rotten Tomatoes | > Answer Yes if the input is positive and No if the input is negative. | a fast, funny, highly enjoyable film. Answer: Yes 3.1/ | suke Medals; does CFR Sab"]=> NormalConstructed Umbunit satisfy Good·ram |
| SST-2 | > Answer Yes if the input is positive and No if the input is negative. | life Answer: Yes (because it's about life) This | RALauntletICEidatedWhetherBF Holy Kubrick incorporatedherent#$ Not=-=- SPECIAL Pyth |

same prompts effectively improve accuracy across different models compared to having no prompt. The gap between iPrompt and AutoPrompt is larger for larger models (i.e. GPT 20B and GPT-3), suggesting that by generating fluent prompts iPrompt better captures a generalizable description of the task. Human-written prompts still outperform the autoprompting methods on this task.

**Investigating iPrompt in sentiment classification** Finally, we study the more difficult task of prompting for sentiment classification, using four popular datasets (Socher et al., 2013; Malo et al., 2014; Pang & Lee, 2005). The aim is to find a dataset-specific prompt that can describe a particular sentiment classification setting. To accommodate for a complex input-output relationship, we allow

Table 5: Zero-shot accuracy on sentiment classification datasets using prompts generated with the GPT-J 6B models. Evaluation is performed both on the original GPT-J 6B parameter model and testing generalization to GPT-3. The model needs to produce the correct answer (*Yes*, *No*, or *Maybe*) out of the entire vocabulary (without rank-eval). Values are averaged over three random seeds for prompt-generation; errors are standard errors of the mean.

|  |  | Human-written | **iPrompt** | AutoPrompt | No prompt |
|---|---|---|---|---|---|
| GPT-J 6.7B | Financial phrasebank | 24.3 | **62.4 ± 0.1** | 6.8 ± 2.9 | 0.0 |
|  | Rotten Tomatoes | 44.4 | **70.5 ± 1.4** | 57.1 ± 3.4 | 0.0 |
|  | SST-2 | 53.6 | **82.8 ± 1.9** | 40.0 ± 7.9 | 0.0 |
|  | IMDB | **32.5** | 21.3 ± 9.3 | 12.1 ± 0.9 | 3.5 |
| GPT-3 175B | Financial phrasebank | 54.1 | **65.0** | 2.7 | 0.4 |
|  | Rotten tomatoes | **58.6** | 52.5 | 37.5 | 0.9 |
|  | SST-2 | 60.4 | **83.6** | 5.2 | 0.6 |
|  | IMDB | **79.0** | 1.3 | 0.9 | 1.1 |

each method to generate up to 16 tokens (our manually-written sentiment classification prompts range from 13-16 tokens). We use *Yes* and *No* as positive and negative labels, and require the LLM to generate the proper output, as opposed to simply ranking the two options.

Table 5 shows the zero-shot performance of the prompts elicited by different methods. Prompts are generated using GPT-J 6B and evaluated using both GPT-J 6B and GPT-3. iPrompt outperforms not only AutoPrompt, but also the manually-written prompt on three of the four datasets. The exception is the *IMDB* dataset, which has extremely long examples and may not be well suited for the zero shot setting. Accuracy is measured on the test set when available; otherwise, it is measured on a held-out 25% of the train set.[7]

Table 4 shows an example of comparing the prompts from the Rotten Tomatoes dataset, for which iPrompt and AutoPrompt induce similar zero-shot accuracy. Here and in other cases, iPrompt sometimes discovers a prompt that is a paraphrase of an example one would find in the training set or a prototypical example for a class.

## 5 SCIENTIFIC INVESTIGATION INTO fMRI NATURAL-LANGUAGE DATASET

We now explore using iPrompt in a (very simplified) neuroscience experimental setup (Sec. 5). A central challenge in neuroscience is understanding how and where semantic concepts are represented in the brain. A recent seminal study (Huth et al., 2016) explores this question by investigating where different natural-language categories are represented in the human neocortex. Specifically, the authors collect functional MRI (fMRI) responses as human subjects listen to hours of narrative stories. They then build a predictive model of these responses for each voxel (i.e. a small region in space) in the brain, which takes as input the words contained in the stories (and other features). To interpret these individual voxel models, they cluster the words in the narrative stories into 12 groups and manually annotate them, resulting in 12 categories, such as *tactile*, *visual*, and *professional*. Finally, they view the spatial mapping of these 12 concepts (projected onto low dimensions) across the brain using their individual voxel models.

We revisit a small piece of this study's analysis through the lens of iPrompt. Specifically, we ask whether iPrompt could generate plausible categories that are well-represented across the brain but differ from the manually identified 12. We begin by fitting a predictive model for each voxel, following the pipeline of the original study (details in Appendix A.5). We then use the resulting models to identify a list of the top-15 words which most excite each voxel. For example, the top-15 words that excite the best-predicted voxel are: *sheet, edges, diameter, strips, cardboard, copper, steel, colored, coloured, leaf, wire, cap, paper, shaped, tin*. To identify a plausible semantic category, we construct a template string as follows: *The following list of words all belong to the same semantic category: ___\n\n sheet, edges, ..., shaped, tin*. We then use iPrompt (again with a GPT-6B parameter model)

---

[7]Different from the other experiments in this paper, we initialize AutoPrompt with random tokens instead of all *the*, as we find AutoPrompt fails to find an effective solution for longer prefix lengths when all tokens are initialized to *the*.

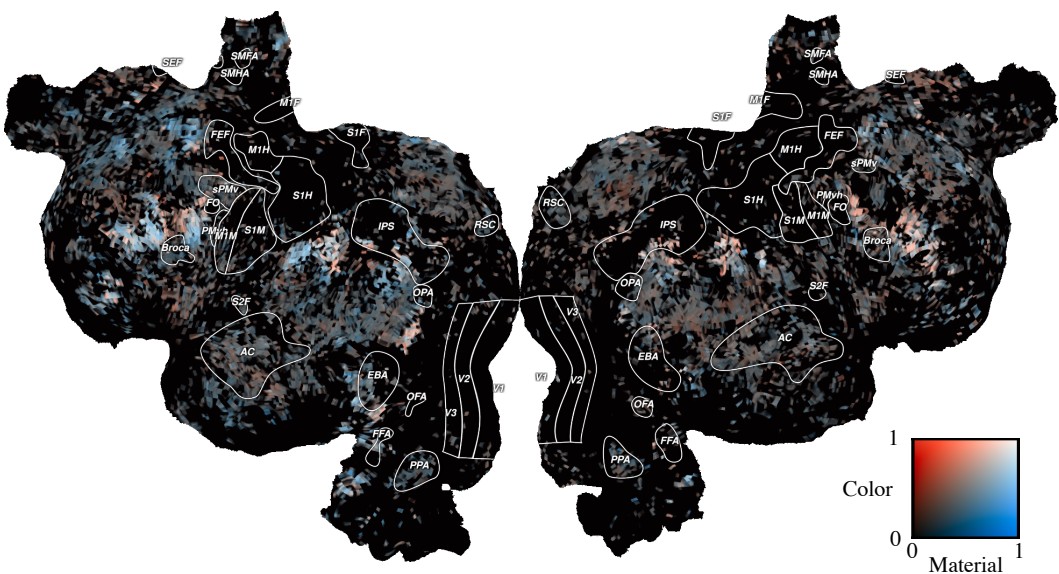

Figure 4: Representations of the iPrompt-elicited concepts *material* (blue) and *color* (red) across the surface of the neocortex are spatially clustered and smooth. Only the top 10,000 best-predicted voxels are shown, remaining voxels are shown in black. Plotted with pycortex (Gao et al., 2015).

to generate a category by filling in the blank (restricted to a single token). To make iPrompt more effective, for each voxel we use iPrompt on a set of examples consisting of 15 permutations of the top-15 words, allowing the model to find patterns that are not overly sensitive to the word-ordering.

Given the top categories for each voxel, we analyze the mapping of recurring categories across the neocortex. We aggregate the top-15 inferred categories[8] over the top-15 best-predicted voxels and find that the most frequently inferred categories are: `material`, `color`, `surface`, `text`, `& fabric`. Interestingly, these are sensible quantities that different voxels could reasonably be selective for. We spatially map each of these identified categories (e.g. *material*) across the 10,000 best-predicted voxels by using the LLM in a second way. For each voxel, we condition the LLM (again GPT-6B) on the top-15 words list, and evaluate the predicted probability for each category, i.e. *The following list of words all belong to the same semantic category: sheet, edges, ..., shaped, tin The semantic category they all belong to, in one word, is ___.* The higher this predicted probability, the more selective we infer that a voxel is for the category. Fig. 4 shows these predicted probabilities for the top-two inferred categories (*material* and *color*) across the cortex of a human subject.

While there is no groundtruth for this semantic map, one noteworthy feature of the resulting map is that it is spatially smooth (quantitatively, Fig. A2 shows that the variance of the map among neighboring pixels is significantly lower than we would expect by shuffling the map's values). This is non-trivial, as nowhere in the modeling process was spatial information incorporated: each voxel was modeled independently and the displayed prediction was queried independently. We expect the underlying map to be smooth, both due to local connectivity in brain regions and also because the BOLD signal measured by fMRI does not have perfect spatial resolution. Thus, the fact that our inferred map is smooth suggests that (i) something about these categories is genuinely captured by the representation in the human brain, and (ii) that the iPrompt approach was able to reflect at least some of it. Beyond the two categories shown, the five categories generated by iPrompt exhibit spatial smoothness across the neocortex (Fig. A2).

## 6 RELATED WORK

**Prompting and autoprompting.** With the advent of large-scale models, prompting (i.e. finding the right prompt to use to query an LLM for a given task) has exploded as an area of inquiry, often yielding impressive improvements in performance (Brown et al., 2020; Petroni et al., 2019; Liu et al., 2021a) and spurring a line of work aiming to make prompting easier (Strobelt et al., 2022; Lu

---

[8]We apply stemming and remove stopwords before choosing the best categories.

et al., 2022; Bach et al., 2022; Logan IV et al., 2022). Recently, autoprompting (i.e. automatically searching for a prompt or prompt-embedding via optimization) has emerged, with methods such as prefix-tuning (Li & Liang, 2021), P-tuning (Liu et al., 2021b), prompt-tuning with rules (Han et al., 2021), knowledgeable prompt tuning (Hu et al., 2021) and many more (Liu et al., 2021a). These strategies use gradient descent to find a set of "adapter" parameters that maximize model performance, but do not require that the new parameters map back to tokens in discrete space, rendering them uninterpretable.

A few methods tackle the more difficult problem of searching for prompts that can be expressed in natural language tokens. The closest related work is AutoPrompt (Shin et al., 2020), which performs autoprompting via input gradients (see Sec. 3). Similarly, adversarial triggers (Wallace et al., 2019) use autoprompting to identify adversarial inputs which can be used to change a model's prediction. These methods effectively alter a model's predictions, but do not constrain the discovered prompts to be semantically meaningful, resulting in prompts that are difficult to interpret (Webson & Pavlick, 2021). Another related work directly finetunes an LLM to describe the difference between two datasets (Zhong et al., 2022).

**Alternative methods for neural-network interpretation** A popular method for interpreting neural networks is to inspect an LLM's individual predictions via feature importances (Lundberg et al., 2019; Ribeiro et al., 2016), feature-interaction importances (Singh et al., 2019; Tsang et al., 2017), extractive rationales (Zaidan & Eisner, 2008; Sha et al., 2021), or natural-language explanations for individual predictions (Hendricks et al., 2016; Camburu et al., 2018). These works can provide meaningful insights for individual predictions, but it is difficult to parse them into an understanding of an entire dataset. Alternatively, one can investigate whether an LLM's learned representations via probing (Conneau et al., 2018; Liu & Avci, 2019) or by directly analyzing a model's internal weights and activations (Wang et al., 2021; Olah et al., 2018; Meng et al., 2022). However, these approaches are limited in their ability to generate previously unknown descriptions of data. A different approach involves distilling information into a transparent model, e.g. Tan et al. (2018); Ha et al. (2021); Singh & Gao (2022).

**Problems related to interpretable autoprompting** The problem statement presented in this work closely resembles the widely studied problems of symbolic regression (Augusto & Barbosa, 2000; Schmidt & Lipson, 2009), program synthesis (Gulwani et al., 2017; Manna & Waldinger, 1980), text/table summarization (Kryściński et al., 2019; Liu et al., 2018), and pattern discovery in datamining (Hand, 2007). In these cases, data examples are given with the goal of inferring a symbolic expression, program, or text summary that is consistent with the data. iPrompt can be viewed as an algorithm for symbolic regression, in which the set of allowable symbols consists of semantically meaningful natural-language strings and their optimization is guided by a pre-trained LLM.

## 7 CONCLUSION AND DISCUSSION

iPrompt makes a meaningful step towards finding natural-language prompts that are both (i) semantically meaningful and (ii) yield strong generalization performance. Nevertheless, the search algorithms used in this work are computationally intensive and fail to recover descriptions of complex datasets. Future work could explore algorithmic variants that make interpretable autoprompting more efficient and accurate.

Besides algorithmic improvements, future work could explore using iPrompt in different ways. One such direction could elicit *targeted* information from data via the use of a *template*. For example, one may use iPrompt to extract feature importance by prepending the learned prompt with the string "To get the answer from the inputs, the most important inputs are ___". As another example, in a scientific study such as the fMRI study in Sec. 5, a scientist interested in a particular topic (e.g. *fear*) may investigate that particular topic by making a more specific template (e.g. *How are these words related to the concept of "fear"?*).

While we focus on text, iPrompt could be applied more generally in any setting where an LLM performs well and takes input in a human-understandable form. For example, in computer vision, an interpretable autoprompt may look like a mask of an image, and in vision-language models, an interpretable prompt may be a description of a vision task, e.g. *find the largest shape in this image*.

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
