# OpenReview forum: "Explaining Patterns in Data  with  Language Models via Interpretable Autoprompting"
_ICLR.cc/2023/Conference — Submitted to ICLR 2023_

### Official Review · Reviewer_a8A9 · 2022-10-24

**Confidence:** 4
**Correctness:** 2
**Technical Novelty And Significance:** 3
**Empirical Novelty And Significance:** 3
**Recommendation:** 3

**Clarity, Quality, Novelty And Reproducibility:**

Clarity and Quality: Overall the paper is well-written. Some details are not clear, e.g., what is $n$ used in experiments.

Novelty: The idea of generating interpretable prompts is somewhat novel.

Reproducibility: code is provided in supplementary materials.


**Strength And Weaknesses:**

Strengths:
- The problem of auto generating prompts/explanations given a dataset is important to the research community on prompting/language models.
- The idea of generating prompt candidates and then reranking/refining them as optimization is somewhat novel.

Weakness:
- The experiment results are relatively weak.
1) the main results are only shown for two synthetic tasks, math and ANLI, and are only compared to two relatively weak baselines, AutoPrompt and suffix decoding. Why there is no human-written baselines? In addition, there are more recent methods that try to find better discrete prompts like:

a) Gao et al. Making pre-trained language models better few-shot learners. ACL 2021.

b) Deng et al. RLprompt: Optimizing discrete text prompts with reinforcement learning. EMNLP 2022.

c) Schick et al. Automatically identifying words that can serve as labels for few-shot text classification. COLING 2020.

which should be cited and compared with if possible.

2) Table 5, on the more realistic task sentiment classification, iPrompt was compared with human-written prompts which shows a relatively better performance. How are the human-written prompts determined? Existing work has shown that instructions/prompts have a large effect on the final task performance, hence for a fair comparison the authors should show best/worst/avg results w.r.t. multiple possible human-written prompts, e.g., from PromptSource (https://github.com/bigscience-workshop/promptsource) where multiple prompts are crowd-sourced for various tasks including sentiment classification.

3) In Table 3, for ANLI, iPrompt is on par with average suffix decoding. Is there any analysis to explain why it can't achieve a better performance on this task?

- All the experiment results are shown on relatively easy tasks (where the prompt can be easily defined/written). Can the authors show results on more complex NLP tasks? E.g., question-answering, natural language inference (the actual task instead of the synthetic one used in the paper). This could better demonstrate how the proposed method is able to generate more complex prompts and understand "what each task is actually doing" rather than just providing label choices.

- Better interpretability of the generated prompts is a major claimed contribution of this paper, but it's not well supported and there is no associated study on this. Can the authors provide a human eval to show the generated prompts indeed have better interpretability (and how close it is to human-written prompts)? Based on Table 4, it's really hard to tell how interpretable the iPrompt-generated prompts are, since many of them do not make much sense to humans, e.g., "Select currency code for a new" (truncated randomly), "What is the opposite of 1" (what is "1"?). Especially for the more realistic task sentiment classification, the prompts generated by iPrompt do not seem meaningful at all.

- Some experimental details are not clear. What is $n$ used in experiments for each task? Is there an ablation study on $n$ to show how many samples are needed for iPrompt to perform well?

**Summary Of The Paper:**

This paper proposes a method called iPrompt to achieve interpretable autoprompting, that iteratively generates possible explanations of the input data, rerank them, and finally generate new explorations via truncating generated candidates and using them as prefixes to generate new candidates.

The authors show experiments over Synthetic Math and ANLI where the proposed approach outperforms AutoPrompt and Average Suffix decoding. On sentiment classification datasets the authors show quite competitive performance of iPrompt compared to human-written prompts.

**Summary Of The Review:**

Overall the problem studies is interesting and the method is somewhat novel, but the experimental results are fairly weak and the interpretability of the generated prompts are not well supported. Thus I recommend rejection.

---

> ### Author Response · Authors · 2022-11-16
> **Response to Reviewer a8A9**
>
> Thank you for your detailed review and extensive feedback! We plan to integrate some of these additional baselines, as discussed below:
>
> > Why are there no human-written baselines [for the prompt recovery task]?
>
> We did not consider providing examples to humans and asking them to write the prompt, but hope to add this baseline initial experiment to a future version of the manuscript.
>
> > How are the human-written prompts determined [for Table 5]?
>
> We manually wrote prompts for each experiment. In the future we will average model performance over a selection of human-written prompts.
>
> > In Table 3, for ANLI, iPrompt is on par with average suffix decoding. Is there any analysis to explain why it can't achieve a better performance on this task?
>
> The ANLI tasks are relatively quite difficult zero-shot for the 6B parameter GPT-J model. We plan to add results on larger, more capable models, which we expect to increase the gap in performance between iPrompt and average suffix.
>
> > Can the authors show results on more complex NLP tasks?
>
> We will add results in the future that search for prompts for more complicated tasks than sentiment analysis.
>
> > Can the authors provide a human eval to show the generated prompts indeed have better interpretability (and how close it is to human-written prompts)?
>
> Human evaluation of prompts is a good idea. We manually verified interpretability of our prompts compared to AutoPrompt, which were not human-readable, let alone fluent. In the future we can perform more thorough human studies.
>
> > Is there an ablation study to show how many samples are needed for iPrompt to perform well?
>
> The appendix shows the efficacy of the best prompts found by iPrompt across training. The number of samples varies widely by dataset and task, but generally a few hundred or a couple thousand data steps of search are required to find a good (human-level prompt) for a large dataset.

---

### Official Review · Reviewer_R7d2 · 2022-10-25

**Confidence:** 4
**Correctness:** 3
**Technical Novelty And Significance:** 3
**Empirical Novelty And Significance:** 2
**Recommendation:** 5

**Clarity, Quality, Novelty And Reproducibility:**

Clarity and Quality: The paper is well-written and has a clear flow.

Novelty: The idea that using the PLM model to generate a prompt by itself is not new. However, the proposed design of the prompt evolution algorithm is interesting.

Reproducibility: Code is provided in the supplementary materails.

**Strength And Weaknesses:**

Strength:
1. The paper is good writing and easy to follow.
2. The proposed method can search readable prompts without access to LLM parameters and gradients.

Weakness:
1. The proxy task(evaluation task) is important for the prompt search, however, the authors did not investigate more on the selection of the proxy task. The current proxy task is to measure the closeness and similarity to the "ground-truth prompt". However, what is the definition of a "ground-truth prompt"? The author did not elaborate more. If it is some existing description of the task label, it may not a good prompt to serve as a "ground-truth" prompt. Did the author explore other evaluation methods to select good prompts?
2. Some results(e.g. Table5) are not convincing. As the PLM zero-shot performance is sensitive to the prompt, the authors should report the average accuracy and std across several prompts when compared with other baselines. The Human-written performance of Rotten Tomatoes/SST-2/IMDB seems too low, as I have simply tried several manual prompts on GPT2-xl(1.3B)  and achieve much higher accuracy. (i.e., the IMDB accuracy of prompt *' The IMDb movie review in negative/positive sentiment is: "<S1>"'*  is 83.4)
3. The compared baseline is weak. More recent baselines should be considered, e.g., Making pre-trained language models better few-shot learners(https://arxiv.org/pdf/2012.15723.pdf)





**Summary Of The Paper:**

This paper proposes a method named **iPrompt**  to automatically generate high-quality prompts using LLM. iPrompt iteratively mutates top prompts by (1) generating explanations with an LLM (2) reranking them based on their performance and (3) exploring new candidates. The authors verify the effectiveness of the proposed method on Math, Sentiment, NLU, and fMRI datasets.

**Summary Of The Review:**

Overall, this paper investigates a practical and interesting direction that generates readable and high-quality prompts by PLM itself. The designed algorithm is well-motivated. However, the experiment part is relatively weak and cannot prove the empirical contribution of this paper.

---

> ### Author Response · Authors · 2022-11-16
> **Response to Reviewer R7d2**
>
> Thanks for the thorough feedback, especially for raising questions about the results in Table 5. We will address the experimental issues and add information about the stopping condition of our algorithm in the final version.
>
> > The proxy task(evaluation task) is important for the prompt search, however, the authors did not investigate more on the selection of the proxy task. … Did the author explore other evaluation methods to select good prompts?
>
> To clarify, we select prompts based on their usefulness for the task, as measured by zero-shot accuracy with the prompt prepended. We evaluate the final prompts based on similarity to the ground-truth prompt and test-set generalization accuracy. We agree that closeness to the ground truth prompt may not be the most important evaluation metric for generated prompts.
>
> > The PLM zero-shot performance is sensitive to the prompt, the authors should report the average accuracy and std across several prompts when compared with other baselines.
>
> In the future we plan to average performance across multiple human-written prompts for each task.
>
> > The Human-written performance of Rotten Tomatoes/SST-2/IMDB seems too low.
>
> The performance is measured based on whether the LLM outputs the correct label token exactly, without any mapping or constraints on the output space. GPT-3 often fails to output a label token (“positive” or “negative”), which is why the baseline is close to 0% instead of 50%.

---

### Official Review · Reviewer_5C3P · 2022-10-25

**Confidence:** 4
**Clarity, Quality, Novelty And Reproducibility:** Please refer to Strength And Weaknesses
**Correctness:** 2
**Technical Novelty And Significance:** 2
**Empirical Novelty And Significance:** 2
**Recommendation:** 3

**Strength And Weaknesses:**

Strength:
1. According to Table 4, the prompts found by iPrompt are interpretable.
2. According to Table 3, the prompts found by iPrompt have higher generalization ability than baselines.

Weak.
1. the problem is not well-defined. In particular, how to assess the interpretability of prompts? The metrics proposed in this paper only consider "whether the generated explanation contains one of a set of problem-specific keywords". I would expect a more reasonable metric.
2. Whether the algorithm is reasonable is not explained. (1) The iteration does not always converge. (2) Why does the algorithm need multiple iterations? I do not understand the necessity of this iteration. (2) How to ensure that the model generates explanations for the relationship between input and output in the proposal step, e.g. in Fig 3, I don't understand why the LLMs generate "combine the numbers".
3. I have doubts about some experiments. I think some baselines are designed incorrectly. For example, (1) why is the accuracy of GPT-3 only 28.4 for the simple Inverse synthetic math task and 0.6% for the SST-2 task? I think the authors deliberately chose the setting that favors their method. (2) AutoPrompt apparently does not have the ability of generalization across models, so it is not suitable as a baseline for Table 3.
(4) The task involved in the experiment is very simple. I think this means that the method proposed in this paper is not suitable for solving complex tasks.

**Summary Of The Paper:**

This paper attempts to find interpretable prompts for the few-shot tasks. An iterative local search algorithm is proposed. The algorithm exploits the ability of LLMs to provide interpretable prompt. The algorithm is evaluated on some simple datasets.


**Summary Of The Review:**

Overall I feel that the problem is not clearly defined. This further led to confusion in the experimental design and inappropriate selection of baselines.

---

> ### Author Response · Authors · 2022-11-16
> **Response to Reviewer 5C3P**
>
> Thank you for the helpful review! We address your questions below:
>
> > How to assess the interpretability of prompts?
>
> The datasets here were deliberately chosen/constructed to make it easy to assess the interpretability of prompts: prompts for Math and NLI can all be verified based on the presence of a single word. We will update the experimental results with a more complicated metric like BLEU score so that our procedure could be used to measure the usefulness of prompts for more complex tasks in the future.
>
> > Whether the algorithm is reasonable is not explained. (1) The iteration does not always converge. (2) Why does the algorithm need multiple iterations?
>
> The prompt may not fully converge due to the difficulty of searching the full space of possible strings, but improvements become marginal after some number of iterations, at which point the search can be stopped.
> iPrompt needs multiple iterations because we find that re-initializing with better prompts significantly improves performance. The appendix contains loss curves for the sentiment analysis tasks which illustrate how both iPrompt and AutoPrompt require many iterations to converge to a useful prompt. We will add additional analysis of prompt-learning dynamics in the future.
>
> > Why is the accuracy of GPT-3 only 28.4 for the simple Inverse synthetic math task and 0.6% for the SST-2 task?
>
> The numbers we provide are GPT-3’s accuracy at the task without any prompt. The inputs come in the format “Input: x \n Output: y”. For math, it is unlikely that GPT-3 will know which mathematical operation to perform without any prompting. For SST-2, GPT-3 outputs one of the label tokens (“positive” or “negative”) just 0.6% of the time.

---

### Author Response · Authors · 2022-11-16
**General Response to Reviewers**

Thanks to all the reviewers for their time and thoughtful comments. We have responded to the reviewer comments individually and have clarified many concerns. We also would like to emphasize that our work focuses on the problem of dataset explanation. While we do have some experiments showing that iPrompt can be used to induce strong generalization accuracy, our main results are on the ability of iPrompt to recover underlying patterns in datasets and express them in fluent natural language.

---

### Decision · Program_Chairs · 2023-01-20

**Decision:**

Reject

**Justification For Why Not Higher Score:**

The reviewers were in agreement that the paper did not meet the threshold for publication.

Quoting: The manuscript would benefit from some additional experiments and baselines along the lines of what has been discussed in review a8A9 and the author's response. The paper would be strengthened by human written baselines, and few shot baselines (for the accuracy score). The paper would also be strengthened by studying behavior across a variety of model scales and understanding the dependence on the number of iPrompt iterations.


**Justification For Why Not Lower Score:**

N/A

**Metareview: Summary, Strengths And Weaknesses:**

This paper studies whether LLMs can understand the semantic content of datasets through the lens of automatic interpretable prompt generation. This is a very compelling application of LLMs and iPrompt is an interesting approach. The evaluation metrics seem reasonable, especially manual inspection – the gold standard for prompt correctness and interpretability. That being said, the manuscript would benefit from some additional experiments and baselines along the lines of what has been discussed in review a8A9 and the author's response. The paper would be strengthened by human written baselines, and few shot baselines (for the accuracy score). The paper would also be strengthened by studying behavior across a variety of model scales (perhaps the OPT model family would provide a natural way to study a range of model scales) and understanding the dependence on the number of iPrompt iterations. Please revise and resubmit as this is interesting work!